# Effect of Intrinsic Tannins on the Fermentation Quality and Associated with the Bacterial and Fungal Community of Sainfoin Silage

**DOI:** 10.3390/microorganisms10050844

**Published:** 2022-04-20

**Authors:** Rongzheng Huang, Fanfan Zhang, Ting Wang, Yulin Zhang, Xiao Li, Yongcheng Chen, Chunhui Ma

**Affiliations:** Grassland Science, School of Animal Technology, Shihezi University, Shihezi 832000, China; huangrz2013@163.com (R.H.); zhangfanfan@shzu.edu.cn (F.Z.); wt18683945459@sina.cn (T.W.); 18890965102zyl@sina.com (Y.Z.); lx744364871@sina.cn (X.L.); cyc159357111@sina.cn (Y.C.)

**Keywords:** sainfoin silage, condensed tannins, bacterial community, fungi community

## Abstract

Sainfoin (*Onobrychis* *viciifolia*) is rich in condensed tannins (CT). CT function includes inhibiting bacterial and fungi activity during the ensiling process. We used polyethylene glycol (PEG) to deactivate tannin activity to find out the effects of CT. The results show that the addition of PEG increased dry-matter loss (8.32% vs. 14.15%, on a dry-matter basis) after 60 d of ensiling, and also increased lactic acid (10.90% vs. 15.90%, on a dry-matter basis) and acetic-acid content (7.32% vs. 13.85%, on a dry-matter basis) after 30 d of ensiling. The PEG-treated group increased its *Pediococcus* relative abundance (0.37–3.38% vs. 7.82–23.5%,) during the ensiling process, increased its *Gibellulopsis* relative abundance after 3 d of ensiling (5.96% vs. 19.52%), increased its *Vishniacozyma* relative abundance after 3 d and 7 d of ensiling (2.36% vs. 17.02%, 3.65% vs. 17.17%), and increased its *Aspergillus* relative abundance after 7 d, 14 d and 60 d of ensiling (0.28% vs. 1.32%, 0.49% vs. 2.84% and 1.74% vs. 7.56%). However, the PEG-treated group decreased its *Alternaria* relative abundance during entire ensiling process (14.00–25.21% vs. 3.33–7.49%). These results suggest that condensed tannins inhibit lactic-acid bacteria fermentation though reducing *Pediococcus* activity, and inhibiting fungi activity depending on different strains.

## 1. Introduction

A large number of agricultural scientists have studied condensed tannins (CT) over more than 50 years. A major focus is interactions of CT with proteins that can generate important agricultural effects, such as protecting forage protein during ensiling and rumen digestion. This protection can improve silage quality and increase the flow of undegraded proteins into the abomasum. CT is also beneficial for the environment, as it leads to reduced greenhouse-gas (methane) emission from ruminants [1]. Various plants contain CT, such as sericea lespedeza (*Lespedeza cuneata* Dum.-Cours.), crown vetch (*Coronilla varia* L.), hedysarum (*Hedysarum alpinum* L.), birdsfoot trefoil (*Lotus corniculatus* L.) and sainfoin (*Onobrychis viciifolia*). Among these plants, CT from sainfoin has the highest capacity to bind protein and causes the least inhibition of cellulose digestion by rumen bacteria [2]. Sainfoin is widely cultivated in the Middle East plateaus, Eastern Europe, Asia and North America because of its high organic-matter content and protein digestibility [3]. Sainfoin has the ability to tolerate drought and alkaline soil [4]. Some of the observed advantages of sainfoin for ruminants include decreased nitrogen (N) excretion in urine and increased fecal N excretion, improved organic-matter (OM) and protein digestion, and increased intestinal amino-acid absorption, which is beneficial for the animals [5,6]. It is well-known that when conserving legumes as silage, severe protein degradation occurs, resulting from the combined action of both plant proteases and microbial activity [7]. Limited proteolysis during sainfoin ensiling was attributed to CT binding with proteins and its antimicrobial activity, which supplied more utilizable CP relative to total CP at the duodenum [8]. Several previous studies showed that CT in sainfoin inhibited *Escherichia. coli* O157:H7 [9,10]; however, this observation depended on CT dose and *E. coli* strain. It is not clear how the CT in sainfoin affect the bacterial and fungi community and the quality of silage. Polyethylene glycol (PEG) is a commonly used CT-inactivating agent that can be used to study the effect of tannin on silage because it forms stable PEG-CT complexes, attributed to hydrogen bonding between PEG and CT [11]. Previous studies showed that the addition of PEG increased soluble N, nonprotein N, lactic acid (LA), and ammonia N (AN) in purple prairie clover (*Dalea purpurea* Vent.) silage, and that CT decreased bacterial and fungi diversity during ensiling [12].

To clarify the bacteria and fungi community during ensiling would illuminate the fermentation process of ensiled forage [13]. As far as we know, however, few researchers have focused on the effects of CT on the bacterial and fungi community, and fermentation characteristics in sainfoin silage. Therefore, the present study was conducted to investigate the effects of CT on the bacterial and fungi community, and fermentation characteristics of ensiled sainfoin.

## 2. Materials and Methods

### 2.1. Forage and Ensiling

Sainfoin (*Onobrychis viciifolia*) was planted in Shihezi City, Xinjiang Province, China. Whole-plant sainfoin was harvested in July 2021 at the early flower stage. After wilting for 12 h to a dry-matter (DM) content of approximately 240 g/kg fresh weight, the sample was chopped into 1–2 cm pieces. The sample was then sprayed with a solution of 640 g/L PEG (Sigma, molecular weight 6000) at A rate of 217 mL/kg DM to achieve a CT:PEG ratio of 1:2 in the samples [12]. Prepare 3 separate piles and treated separately, bagged randomly for control and PEG treated groups.The control samples were sprayed with an equivalent amount of distilled water. After spraying, 1000 g samples of both the treated and control chopped sainfoin were packed into a polyethylene plastic bags (30 cm × 50 cm), then compacted and sealed using a vacuum sealer. Three replicates were prepared of the control and treated samples. The bags were stored indoors at 23 °C. Samples were taken from wilted and ensiled sainfoin (days 3, 7, 14, 30 and 60 of silage fermentation) for later analysis.

### 2.2. Characteristics of Wilted and Ensiled Sainfoin

Silage samples from a separate bag for a total 5(five fermentation days) *2(two treat-ment) *3(replicates) of each silage were collected. Silage samples were dried and ground, using 1.0 mm in preparation for obtaining the DM content. The water-soluble carbohydrates (WSC) were determined by using boiling water for extraction according to the anthrone method [14]. Total nitrogen (TN) was determined on an automatic Kjeldahl nitrogen analyzer (K9840, Hanon Co., Ltd., Qingdao, China) according to the procedure described by Association of Official Agricultural Chemists. Neutral detergent fiber (NDF) and acid detergent fiber (ADF) contents were analyzed using methods previously described [15]. 

For fermentation characteristics, the sample from each different fermentation day and treated (control and PEG) silage was taken at 20 g, then combined with 180 mL distilled water and thoroughly blended in a homogenizer (L-1BA, Kuansons Biotechnology Co., Ltd., Shanghai, China). After filtering the mixture of sample solution by four layers of cheesecloth, the supernatant was collected for volatile fatty acids (VFA) and AN content analysis. The pH was measured using a portable pH meter (PHS-3C, Instrument and Electrical Science Instrument Co., Ltd., Shanghai, China). AN content was determined by the phenol-hypochlorite colorimetric method [16]. For measuring the volatile fatty-acid content, the supernatant was filtered through a 0.22 μm dialyzer, then analyzed on a 1200 series high-performance liquid chromatography (HPLC) system (Agilent Technologies, Inc., Waldbronn, Germany), using a C18 column (150 mm × 4.6 mm, FMF-5559-EONU, FLM Scientific Instrument Co., Ltd., Guangzhou, China). For the analysis, Na_2_HPO_4_ (1 mM) was used for the mobile phase with a flow rate of 0.6 mL·min^−1^, with an injection volume of 20 μL and oven temperature of 50 °C [4]. The concentrations of extractable, protein-bound, and fiber-bound CT in sainfoin before and after ensiling were analyzed, using CT purified from whole sainfoin plants as a standard, according to previously described methods [17].

Microbial count was carried out according to methods previously described [18]. Briefly, samples were homogenized in 90 mL of sterilized saline, and then the supernatant was serially diluted. Lactic-acid bacteria (LAB) were detected on De Man Rogosa Sharpe (MRS) agar incubating at 37 °C for 48–72 h. Aerobic bacteria were detected on nutrient agar after incubating at 30 °C for 24 h. Yeasts and molds were detected on Rose Bengal agar after incubating at 30 °C for 78–120 h. The microbial data concentrations presented as log-transformed before statistical analysis.

### 2.3. Sequencing Analysis of the Bacterial Community

Total DNA of each sample was extracted with a commercial DNA Kit (FastDNA^®^ Spin Kit for Soil, MP Biomedicals, New York, NY, USA). Primers targeting the V3-V4 regions of 16S rDNA (338F: ACTCCTACGGGAGGCAGCAG; 806R: GGACTACHVGGGTWTCTAAT) and primers targeting ITS1 regions (ITS1F: 5′-CTTGGTCATTTAGAGGAAGTAA-3′; ITS2R: 5′-GCTGCGTTCTTCATCGATGC-3′) of fungi were used to conduct PCR amplification according to [19,20]. The amplicons were extracted, purified and analyzed following the method described by [21]. Three replicates for each sample were conducted, and a mixture of the three replicates for each sample was sequenced. 

### 2.4. Statistical Analysis

The fermentation characteristics of sainfoin silage data were subjected to Two-way ANOVA, 2 × 5 factorial Complete Randomized design (PEG and control treatments × five ensiling-time days). Data were analyzed using SPSS software (IBM SPSS 22.0 Software, New York, NY, USA). Significant differences between each treatment were determined by Tukey’s test when *p* < 0.05. The sequencing data of bacterial communities were analyzed using the online Majorbio Cloud Platform, provided by Majorbio Bio-pharm Technology Corporation, Shanghai, China (www.majorbio.com, accessed on 1 February 2022).

## 3. Results

### 3.1. Characteristics of Wilted and Ensiling of Sainfoin

The DM, CP and WSC content of wilted sainfoin was 236.50 g/kg, 223.12 g/ kg DM and 101.43 g/kg DM, respectively (Table 1).

The characteristics of sainfoin silages are shown in Table 2. 

The extractable CT in sainfoin dropped to 1.96% (on a dry-matter basis) after 3 d of ensiling, and then maintained a constant level during the ensiling process. The protein-bound tannins increased to 6.97% after 3 d of ensiling, and then maintained a constant level during the ensiling process. After 60 d of ensiling, DM losses were 8.32% and 14.15%, respectively. The highest DM loss of the PEG-treated group was observed compared to the control group during the entire ensiling (*p* < 0.05). AN content showed the same results as DM loss during ensiling, which were 1.60 g/kg DM and 2.64 g/kg DM after 60 d of ensiling, respectively. The pH dropped to 4.3 for both the control and PEG-treated groups after 60 d of ensiling. There was no difference in pH between the control and PEG-treated groups during the entire ensiling (*p* > 0.05). The significant difference was found only in LA and acetic acid (AA) content between the control and PEG-treated groups after 30 d of ensiling (*p* < 0.05), which were 108.99 g/kg vs. 158.89 g/kg and 73.15 g/kg vs. 138.47 g/kg, respectively. The AA content in both treatments was high (>10%, on a dry-matter basis) after 60 d of ensiling. There was no difference of LA to AA ratio between control and PEG-treated groups during the entire ensiling (*p* > 0.05), and this ratio was below 3 during ensiling. The LAB counts were highest in both the control and PEG-treated groups after 3 d of ensiling, which were 9.04 Log_10_ cfu/g FM and 9.19 Log_10_ cfu/g, respectively. The PEG-treated group showed a higher trend in LAB count compared with the control group during ensiling, and this trend became a significant difference after 7 d, 30 d and 60 d of ensiling (*p* < 0.05). The mold count had the same trend as the LAB count during ensiling, which showed a significant difference after 3 d, 30 d and 60 d of ensiling (*p* < 0.05). The yeast count showed significant difference between the control and PEG-treated groups after 3 d of ensiling (*p* < 0.05), which were 3.82 Log_10_ cfu/g FM and 4.33 Log_10_ cfu/g FM, respectively.

### 3.2. Bacterial Community of Sainfoin Silage

Alpha diversity of the bacterial community for sainfoin silage is shown in Table 3. 

The Good’s coverage index was higher than 0.99, indicating that the degree of sequencing was sufficient for the bacterial community analysis. With or without PEG, the richness of the microbial community in sainfoin decreased during ensiling. The richness and diversity of the microbial community increased in sainfoin with PEG treatment at 30 d. However, there was no difference in the microbial community richness and diversity between silage with or without PEG at 60 d of ensiling. In addition, using a weighted uniFrac distance to assess the bacterial communities, a clear distinction was found between the bacterial communities in the control and PEG-treated silage at 30 d of ensiling (Figure 1; *p* < 0.01; R = 0.7587). 

This may indicate that the impact of tannins on microbial activity at 60 d is not pronounced, likely because the diversity of the microbial community decreased with the increase in the number of dominant bacteria.

The bacterial community of sainfoin at the phylum level before and after ensiling is shown in Figure 2.

In the wilted sainfoin, *Cyanobacteria* (87.54%) was the predominant phylum, followed by *Proteobacteria* (9.85%), *Firmicutes* (1.2%), and *Actinobacteria* (1.05%). 

Analyzing the bacterial community on the genus level (Figure 3) showed that there was a significant difference between the control and PEG silages. In general, *Weissella*, *Lactobacillus*, *Enterobacter* and *Pediococcus* were the four richest bacterial communities on the genus level. During the ensiling process, the *Lactobacillus* relative abundance increased as the days of ensiling increased. The *Lactobacillus* relative abundance of the control and PEG groups was 42.06–90.56% and 27.4–69.92%, respectively. The *Enterobacteria* relative abundance decreased with ensiling days, with a relative abundance for the control and PEG groups at 1.04–11.92% and 2.78–12.66%, respectively. The *Weissella* relative abundance slightly decreased after 7 d of ensiling, then stabilized until 30 d of ensiling in the control silage. After 60 d of ensiling, the *Weissella* relative abundance dropped considerably to 3.85% in the control silage. The *Weissella* relative abundance in the PEG silage decreased slightly for 7 d but stabilized for the remaining ensiling. *Pediococcus* showed the most significant difference in relative abundance of the LAB bacteria (*p* < 0.05), with the *Pediococcus* relative abundance for the control and PEG group at 0.37–3.38% and 7.82–23.5%, respectively.

*Lactobacillus* was the main genus (42.06%) in the control silages at 3 d of ensiling, followed by *Weissella* (31.83%), and *Enterobacter* (17.96%), with the *Pediococcus* relative abundance being the lowest (0.50%). For the PEG group, *Weissella* became the main genus in silages at 3 d of ensiling (42.57%), followed by *Lactobacillus* (27.4%), *Enterobacter* (12.66%) and *Pediococcus* (7.92%). However, there was no significant difference (*p* > 0.05) between these four bacteria on the genus level between the control and PEG silage at 3 d of ensiling. During ensiling, a significant difference (*p* < 0.05) in *Pediococcus* relative abundance between control and PEG silages was found after 7 d of ensiling, with the control and PEG silage *Pediococcus* relative abundance at 0.37% and 14.2%, respectively. This same significant difference for *Pediococcus* relative abundance between control and PEG silages was no longer present at 60 d of ensiling. This suggests that CT affected the activity of *Pediococcus* primarily in the early and middle ensiling stages. 

### 3.3. Fungi Community of Sainfoin Ensiled with or without PEG

Alpha diversity of fungi community show in Table 4. 

As a result, the richness of the microbial community decreased in both the control and treated groups with prolonged ensiling. Addition with PEG, however, increased (*p* < 0.05) the diversity of the microbial community after 3 days of ensiling. There was no difference (*p* > 0.05) in microbial-community diversity between the control and PEG treatments during ensiling days 7 to ensiling days 60. Addition with PEG decreased (*p* < 0.05) the richness of the microbial community during 30 days of ensiling. In addition, weighted UniFrac distance was used to assess fungi communities, as shown in Figure 4. 

The fungi community of sainfoin at phylum level before and after ensiling are shown in Figure 5.

*Ascomycoto* were dominant in all group silages with a relative abundance of 59.51–89.43%, followed by Basidiomycota (7.78–39.18%).

Analysis of the fungi community on a genus level is shown in Figure 6.

*Cladosporium* dominated in wilted sainfoin with a relative abundance of 34.81%, followed by *Filobasidium* (21.28%), *Alternaria* (12.39%), *Gibberella* (11.55%), *Gibellulopsis* (7.95%) and *Vishniacozyma* (4.95%). *Cladosporium* dominated in control groups during entire ensiling with a relative abundance of 20.68–41.30%, followed by *Alternaria* (14.00–25.21%). *Cladosporium* dominated in PEG groups during 30 days of ensiling. Addition with PEG increased *Gibellulopsis* relative abundance after 3 days of ensiling (5.96% vs. 19.52%) and *Vishniacozyma* relative abundance after 3 days and 7 days of ensiling (2.36% vs. 17.02%, 3.65–17.17%). Addition with PEG increased *Aspergillus* relative abundance after 7 days, 14 days and 60 days of ensiling (0.28% vs. 1.32%,0.49% vs. 2.84% and 1.74% vs. 7.56%). Addition with PEG decreased *Alternaria* relative abundance during 60 days of ensiling (14.00–25.21% vs. 3.33–7.49%), *Gibberella* relative abundance after 3 days and 7 days of ensiling (10.07% vs. 3.22% and 15.05% vs. 3.77%) and *Cladosporium* relative abundance after 3 days of ensiling (41.30% vs. 22.05%).

## 4. Discussion

Levels of CT greater than 6% have shown a strong ability to inhibit bacterial activity and protein degradation in silage [18]. In addition, tannic acid is a type of polyphenol substance analogous to CT (both are a complex group of polyphenolic compounds), and 2% tannic acid can inhibit activity in certain bacteria, including lactic acid and coliform bacteria, and decreased protein degradation during ensiling [22]. The CT content-level results suggest that there is biological activity of CT in sainfoin materials. The epiphytic microorganisms and WSC are both important factors for silage fermentation. A WSC content of at least 60 g·kg^−1^ DM is the common threshold level for achieving well-preserved silage [14]. In this study, the WSC content was 101.43 g·kg^−1^ DM, which is sufficient for fermentation. LAB counts in the silage were higher than those of the spoilage organisms in wilted sainfoin, such as molds and yeasts, which might be favorable to the establishment of a dominant LAB community during fermentation.

Previous studies found that protein-bound CT increased because the extractable CT decreased after ensiling [23]. The present study found that extractable tannins bound more protein, then stabilized quickly in the early stage of ensiling. The protein-bound CT stabilized because of pH; CT bound to fraction 1 leaf protein from alfalfa stabilized at pH 4–6 [11]. In the current study, the pH ranged from 4.32–5.41 during the ensiling process. The LA and AA ratio indicates fermentation type: a high LA-to-AA ratio (>3) indicates homolactic fermentation, whereas a lower LA-to-AA ratio (<3) indicates heterolactic fermentation [24]. The control and PEG-treated groups LA-to-AA ratio (<3) during the entire ensiling process suggested a heterolactic fermentation. Notably, however, there was no difference in pH between the control and PEG-treated silages during the entire ensiling process. The higher buffering capacity for legumes (500–550 mE/kg of DM for legumes) was likely the reason for these results [25]. Legume silages generally have a final pH of 4.4–4.5 [26]; in this study, pH was 4.3 in both groups after 60 d of ensiling, which indicates adequate quality for silage. The addition of PEG significantly increased DM loss during the entire ensiling process. These results suggest that PEG likely enhanced the activity of silage microbes. Consistently, PEG significantly increased LA and AA contents, as well as LAB counts after 30 d of ensiling. WSC is the main substrate for LAB fermentation. WSC content dropped 60% of from the initial content after 3 d for both control and PEG-treated groups, suggesting that the raw material had a sufficient LAB number for fermentation. The WSC content were lowest in PEG-treated silages during the entire ensiling, likely resulting from higher LAB activity in treated silage compared to control. LAB converted WSC into organic acids, rapidly reducing pH and preserving silages [14]. These results suggest that CT inhibits LAB activity during ensiling. The antibacterial mechanisms of catechins can be broadly classified into inhibition of virulence factors (toxins and extracellular matrix), cell-wall and cell-membrane disruption, inhibition of intracellular enzymes, oxidative stress, DNA damage and iron chelation of six groups [27]. Four types of catechins are the main components of CT in sainfoin [6], suggesting that CT likely has the same antibacterial mechanisms. Several studies confirmed that CT from sainfoin inhibited bacterial via inhibition of intracellular enzymes, and disrupted cell-wall and cell-membrane integrity [10,28].

Yeast activity in silage usually causes nutrient loss during ensiling. Yeast activity is enhanced by ensiling during the beginning stage due to the aerobic environment, then decreased as the silage pH and oxygen is reduced. AA is recognized as an important yeast inhibitor. Here, yeast counts decreased with the accumulation of AA during the ensiling process for both silage groups. Similar levels of AA were found in the control and PEG silages for the duration of the ensiling process, except at 30 d of fermentation. However, a significant difference between yeast counts was only found at 3 d of fermentation, when control and PEG silage yeast counts were 3.82 and 4.33 log_10_ CFU g^−1^ FM, respectively. The results suggest that AA was not the sole inhibitor of yeast in these silages. Studies have found that CT-inhibited yeast grow during the ensiling process in a species-dependent manner [12]. It can therefore be concluded that CT and AA both affect the yeast population in silage.

After 60 d of ensiling, *Firmicutes* became the dominant phylum in both groups, with a relative abundance of 94.52–97.97%; however, the *Cyanobacteria* relative abundance dropped to 0.12–0.49%. Similar results were observed in [18], except in this study no difference of bacterial relative abundance was found at the phylum level between the control and PEG groups. Such a discrepancy in different studies indicates that there are differences in the bacterial succession in silages using different raw materials, even if the raw materials contain the same type of bacterial communities.

*Lactobacillus* were the main genus in the control and PEG groups during ensiling. The *Lactobacillus* relative abundance (90.56%) was much higher after 60 d of ensiling in the control silages. This is because other bacteria in the community dropped to their lowest level at this stage. Some species of *Lactobacillus*, such as *Lactobacillus plantarum NBRC15891*, *Lactobacillus fermentum NBRC15885* and *Lactobacillus delbrueckii NBRC3073* have a strong tolerance for catechins [29]. Since CT in sainfoin contain four types of catechins, this suggest that the CT in sainfoin have little impact on *Lactobacillus* activity in the ensiling process.

The various *Pediococcus* are facultative heterofermentative LAB, and produce LA and AA by fermenting hexoses and pentoses, respectively [30]. In this study, PEG-treated silage resulted in a significantly higher relative abundance of *Pediococcus* than the control silages from 7 d to 30 d of ensiling (*p* < 0.05). Similar results were found with purple prairie clover silage with added PEG [12]. Some strains of *Pediococcus* species isolated from forage can grow in a 3.5–6.0 pH environment [31]. Furthermore, *Pediococcus* species growth and proliferation are well under a pH range of 4.3–4.9 in sainfoin and alfalfa-mixture silages [32]. In this study, the pH ranged from 4.34–4.8 during ensiling. These results indicate that the *pediococcus* relative abundance may not be affected by pH during ensiling. During the ensiling process, acidity, substrate availability, aerobiosis, moisture, or temperature could affect the fermentation and LAB succession during the ensiling process [33]. In this study, there was no difference in pH between the control and PEG silages. After 60 d of fermentation, the WSC consumption for the control and PEG silages was 69.35% and 76.99% of the initial content, respectively. This observation suggested that the ensiling process was not substrate-limited. Hence, CT strongly influenced LAB succession, particularly for *Pediococcus*.

*Enterobacteria* relative abundance decreased with increasing ensiling days, consistent with the pH decrease from 5.2 to 4.3. The critical pH for *Enterobacteria* was 4.5 [34]. Once the pH dropped below 4.5, *Enterobacteria* were inhibited [35]. The PEG-treated group had no effect on *Enterobacteria* relative abundance compared to control silages. However, the lowest AN level was found in control silage throughout the silage-fermentation process. This seems to disagree with the general assumption that the production of AN in silage usually results from the activity of *Enterobacteria* if *Clostridia* are not detected (in the present study, *Clostridia* was not detected) [25]. In addition, as plant tissue gradually breaks down in prolonged silage and releases intracellular enzymes, or as the pH condition became more favorable to the enzymes, plant proteases have an increased effect on protein degradation [18]. Thus, AN content may mainly come from protein degradation by plant proteases.

*Fusarium* and *Alternaria* are often considered as field fungi; *Penicillium* and *Aspergillus* are storage fungi. Those fungi are usually considered toxin producers. *Fusarium*, *Alternaria* and *Cladosporium* can produce mycotoxins during their growth by infections of grass, resulting in forage crops already containing toxins before being ensiled or grazed [36]; however, very few studies on detection and/or isolation of *Cladosporium* species from silages [37]. The *Cladosporium* were dominant fungi in sainfoin silages, which is in agreement with others observed [12,37]. *Gibberella* is a sexual form of *Fusarium*. *Fusarium* is an important toxin-producing fungus; however, several studies observed that mycotoxins produced by *Fusarium* were already present at the harvest and contents were stabled in silage [38,39]. Furthermore, the main mycotoxin from *Fusarium* is trichothecenes, which is involved in the plant-infection process. Thus, these traits are not needed for *Fusarium* growth on harvested silage [40]. As results indicate, addition with PEG decreased *Gibberlla* relative abundance; however, those impacts may not affect mycotoxin contents in silage. *Vishniacozyma* exists in plants such as alfalfa, maize and wheat crops [41]. The present study also found *Vishniacozyma* activity during 30 d of ensiling.

In general, field fungi activity decreased with prolonged ensiling; however, storage fungi activity increased during ensiled. A previous study observed that *Alternaria* relative abundance decreased; however, *Aspergillus* relative abundance increased in maize silage [42]. The present study showed the same results when the effect of ensiling days was considered. Significantly, there was the lowest relative abundance of *Alternaria* in PEG treatment during the entire ensiling. The results suggested that CT probably promotes *Alternaria* activity during sainfoin ensiling. However, the mechanism of CT promoting *Alternaria* activity during sainfoin ensiling needs further study. As a storage fungus, *Aspergillus* can grow under different environments of silage, such as lower oxygen and pH levels [43,44]. A previous study (in vitro) showed that CT extract from *Mimosa tenuiflora* (Willd.) bark can inhibit *Aspergillus flavus* growth. The present study showed that relative abundance increased in both groups with prolonged ensiling; however, addition with PEG increased its relative abundance after 7, 14 and 60 days of ensiling. The results suggested that CT might inhibit *Aspergillus* growth in sainfoin silages.

## 5. Conclusions

The CT in sainfoin significantly decreased DM loss, content of LA, AA and AN during ensiling. At the same time, CT inhibited bacterial richness and diversity, particularly for *Pediococcus*. CTs in sainfoin have little inhibitive effect on *Vishniacozyma* activity; however, CT showed strong inhibit effect on *Aspergillus* activity during sainfoin ensiling. On the contrary, CTs have no inhibit effect on *Alternaria* activity, but CT probably promotes its growth. This study indicated that CT from sainfoin can inhibit LAB fermentation by reducing *Pediococcus* activity and inhibiting fungi activity, depending on different strains.

## Figures and Tables

**Figure 1 microorganisms-10-00844-f001:**
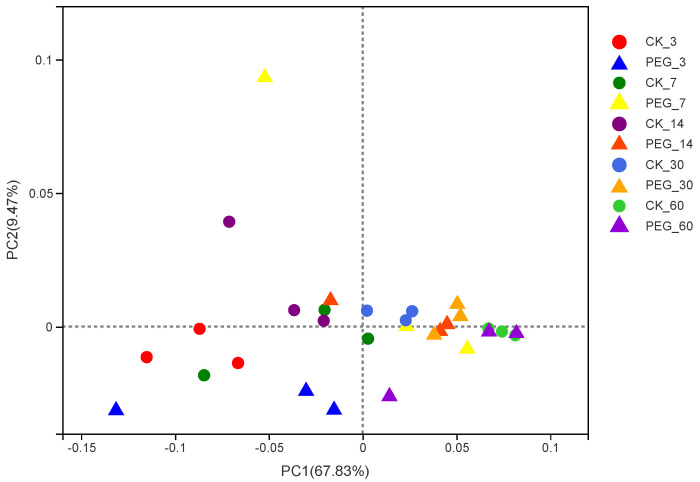
Principal−coordinate analysis (PCoA) plots based on weighted UniFrac distance for bacterial community of treatments × days (CK_3: on day 3 of ensiling without addition of polyethylene glycol (PEG); PEG_3: on day 3 of ensiling addition of PEG, same as others).

**Figure 2 microorganisms-10-00844-f002:**
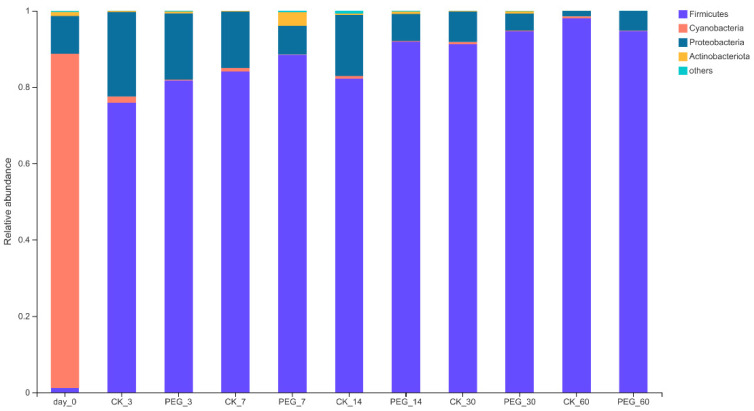
Relative abundance of bacterial community on the phylum level of PEG-treated and control group of sainfoin silage (CK_3: on day 3 of control ensiling; PEG_3: on day 3 of PEG-treated ensiling, same as others).

**Figure 3 microorganisms-10-00844-f003:**
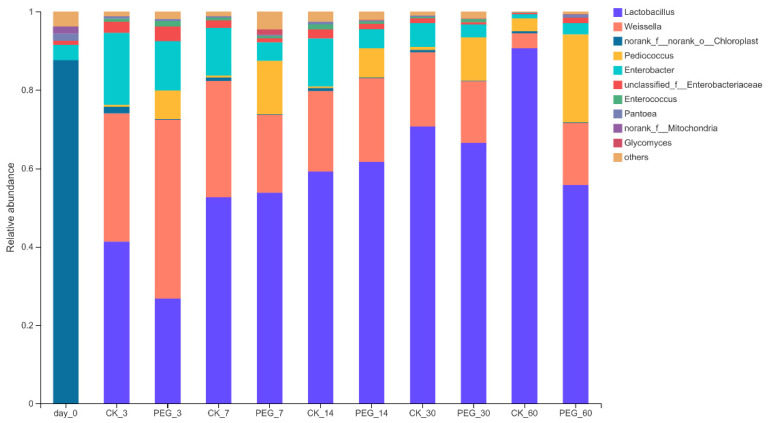
Relative abundance of bacterial community on the genus level of PEG-treated and control group of sainfoin silage (CK_3: on day 3 of control ensiling; PEG_3: on day 3 of PEG-treated ensiling, same as others).

**Figure 4 microorganisms-10-00844-f004:**
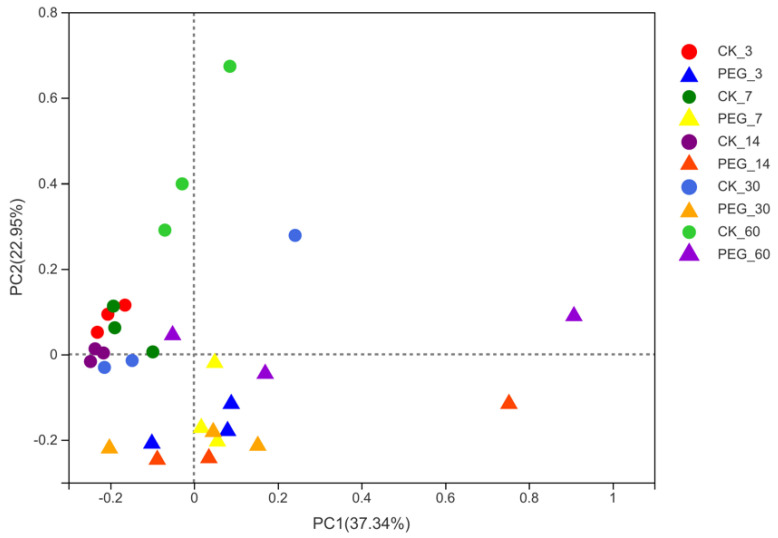
Principal−coordinate analysis (PCoA) plots based on weighted UniFrac distance for fungi community of treatments × days (CK_3: on day 3 of ensiling without addition of polyethylene glycol (PEG); PEG_3: on day 3 of ensiling addition of PEG, same as others) observed a clear distinction (R = 0.3305, *p* < 0.05) between fungi communities in control and PEG-treated silages during the entire ensiling day. These indicate that condensed tannins inhibit fungi activity; however, this effect might quickly disappear with prolonged ensiling.

**Figure 5 microorganisms-10-00844-f005:**
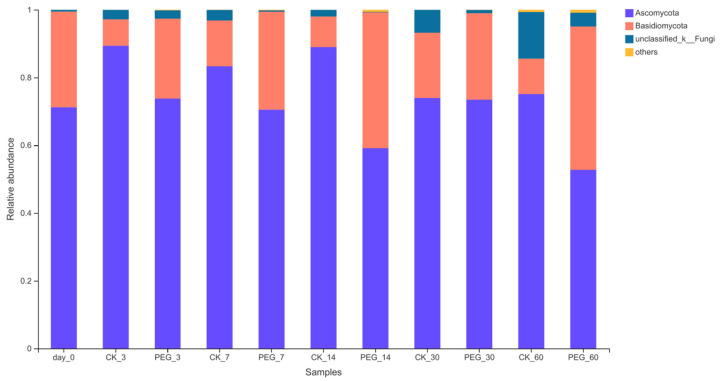
Relative abundance of fungi community on phylum level of PEG-treated and control group of sainfoin silage (CK_3: on day 3 of control ensiling; PEG_3: on day 3 of PEG-treated ensiling, same as others).

**Figure 6 microorganisms-10-00844-f006:**
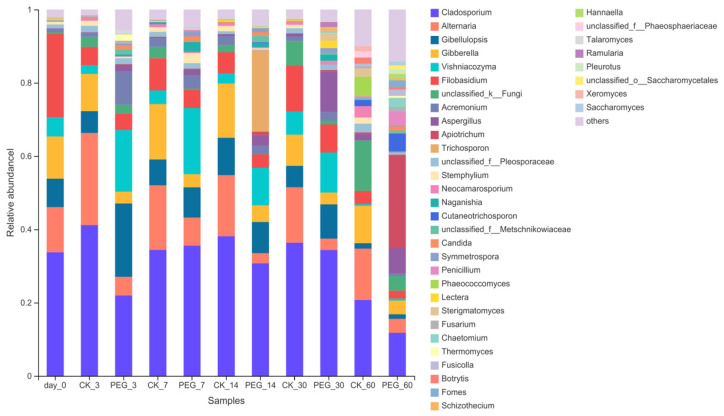
Relative abundance of fungi community on genus level of PEG-treated and control group of ensiled sainfoin (CK_3: on day 3 of control ensiling; PEG_3: on day 3 of PEG-treated ensiling, same as others).

**Table 1 microorganisms-10-00844-t001:** Chemical compositions of wilted whole-plant sainfoin harvested at early flower stage for silage ^1^.

Chemical Composition	Mean ± SE
Dry matter (g/kg fresh forage)	236.5 ± 2.5
Crude protein	223.12 ± 0.61
Water-soluble carbohydrates	101.43 ± 0.89
Neutral detergent fiber	505.83 ± 5.03
Acid detergent fiber	300.66 ± 10.58
Neutral detergent—insoluble protein	129.16 ± 1.11
Acid detergent—insoluble protein	30.23 ± 0.50
pH	5.41 ± 0.01
Soluble protein	10.75 ± 0.24
Extractable condensed tannins (g/kg DM)	47.31 ± 2.46
Protein-bound condensed tannins (g/kg DM)	45.67 ± 1.91
Fiber-bound condensed tannins (g/kg DM)	16.80 ± 0.34
Total condensed tannins	109.78 ± 3.52
Lactic-acid bacteria (Log_10_ cfu/g FM)	6.75 ± 0.06
Yeasts (Log_10_ cfu/g FM)	ND
Molds (Log_10_ cfu/g FM)	5.70 ± 0.04
Aerobic bacteria (Log_10_ cfu/g FM)	7.11 ± 0.03

^1^ *n* = 3. Composition measures are in grams per kilogram of DM, unless otherwise stated. CT content in wilted sainfoin was 109.78 g/kg DM consisting of extractable CT, protein-bound CT and fiber-bound CT, which were 47.31 g/kg DM, 45.67 g/kg DM and 16.80 g/kg DM, respectively. LAB count (6.75 Log_10_ cfu/g FM) was higher than those of undesirable microbes such as molds (5.70 Log_10_ cfu/g FM) and yeasts (not detected).

**Table 2 microorganisms-10-00844-t002:** Ensiling characteristics of sainfoin with or without the addition of PEG ^1^.

Item	Treatment	Days of Ensiling					SEM	*p*-Value		
		3	7	14	30	60		Day	PEG	D*P
DM	CK	22.13Aa	21.60Aa	21.83Aa	21.48Aa	21.68Aa	0.139	<0.01	<0.01	0.173
	PEG	24.16Ba	23.28Ba	23.73Ba	23.46Ba	22.68Bb				
DML (%DM)	CK	6.43Aa	8.67Aa	7.72Aa	9.19Aa	8.32Ba	0.403	<0.01	<0.01	0.299
	PEG	8.57Ab	11.92Aa	10.21Aa	11.19Aa	14.15Aa				
CP (g/kg DM)	CK	240.13Aa	239.64Aa	240.85Aa	244.70Aa	246.79Aa	2.72	<0.01	<0.01	<0.01
	PEG	219.60Ba	202.90Bb	204.52Bb	216.02Bac	208.02Bbc				
NDF (g/kg DM)	CK	493.37Aa	473.01Aa	426.64Aa	448.55Aa	455.10Aa	8.54	<0.01	<0.01	0.540
	PEG	420.10Aa	351.96Bb	348.77Ab	350.79Bb	348.58Bb				
ADF (g/kg DM)	CK	299.39Ab	271.07Ab	322.29Aa	300.73Ab	274.06Ab	4.90	0.105	<0.01	0.011
	PEG	280.77Aa	265.25Aa	239.75Ba	248.98Aa	245.87Aa				
E-CT (g/kg DM)	CK	19.63	20.93	19.38	18.67	16.83	0.721	0.484	-	-
P-CT (g/kg DM)	CK	69.69	64.82	69.73	72.57	69.92	1.03	0.280	-	-
F-CT (g/kg DM)	CK	27.40	25.61	27.14	26.12	25.48	0.407	0.441	-	-
pH	CK	4.61Ab	4.75Aa	4.67Aab	4.57Ab	4.32Ac	0.025	<0.01	0.53	0.064
	PEG	4.56Ab	4.68Aa	4.69Aab	4.60Aa	4.36Ac				
AN (g/kg DM)	CK	0.64Bc	0.94Bb	1.13Bb	1.42Ba	1.60Ba	0.064	<0.01	<0.01	<0.01
	PEG	1.16Ad	1.24Ad	1.51Ac	1.93Ab	2.44Aa				
WSC (g/kg DM)	CK	46.02Aab	41.77Aab	37.95Ab	35.76Abc	31.09Ac	0.89	<0.01	<0.01	<0.01
	PEG	30.89Bab	32.86Ba	30.64Bab	26.47Bbc	23.34Bc				
LA (g/kg DM)	CK	107.74Aa	96.25Aa	86.33Aa	108.99Ba	115.36Aa	4.37	0.003	<0.01	0.063
	PEG	102.72Ab	123.03Ab	127.94Ab	158.89Aa	162.98Aa				
AA (g/kg DM)	CK	67.57Ab	76.66Aab	66.20Ab	73.15Bb	101.62Aa	3.56	<0.01	<0.01	<0.01
	PEG	63.33Ab	80.20Ab	68.61Ab	138.47Aa	117.85Aa				
PA (g/kg DM)	CK	4.72Aa	1.47Ab	0.71Ab	1.41Ab	1.82Bb	0.293	<0.01	<0.01	0.135
	PEG	6.10Aa	1.89Ab	2.28Ab	2.75Ab	5.50Aa				
LA to AA Ratio	CK	1.60Aa	1.26 Aa	1.35 Aa	1.50 Aa	1.13 Aa	0.044	0.014	0.035	<0.01
	PEG	1.61 Aa	1.56 Aa	1.86 Aa	1.15 Aa	1.45 Aa				
LAB Log10 cfu/g FM	CK	9.04Aa	8.82Bab	8.70Aab	8.06Bc	7.13Bd	0.089	<0.01	<0.01	0.068
	PEG	9.19Aa	9.28Aa	8.99Aa	8.60Ab	7.72Ac				
Molds Log10 cfu/g FM	CK	5.06Ba	4.92Aa	4.82Bab	4.78Aab	4.44Bb	0.042	<0.01	<0.01	0.300
	PEG	5.50Aa	5.21Aab	5.20Aab	5.14Aab	5.08Ab				
Yeasts Log10 cfu/g FM	CK	3.94Aa	3.82Ba	3.87Aa	2.85Ab	2.79Ab	0.072	<0.01	0.195	<0.01
	PEG	3.53Abc	4.33Aa	3.78Ab	3.11Acd	2.91Ad				
AB Log10 cfu/g FM	CK	9.15Aa	8.80Bab	8.64Ab	8.15Bc	7.24Bd	0.086	<0.01	<0.01	0.106
	PEG	9.36Aa	9.28Aab	8.95Ab	8.62Ac	7.83Ad				

^1^ *n* = 3.CK: control group; PEG: polyethylene glycol; DM: dry matter; DML: dry-matter loss; CP: crude protein; NDF: neutral detergent fiber; ADF: acid detergent fiber; E-CT: extractable condensed tannins; P-CT: protein-bound condensed tannin; F-CT: fiber-bound condensed tannin; T-CT: total condensed tannin; WSC: water-soluble carbohydrates; AN: ammonia nitrogen; LA: lactic acid; AA: acetic acid; PA: propionic acid; LAB: lactic-acid bacteria; AB: aerobic bacteria. Different capital letters in the same column indicate significant difference (*p* < 0.05). Different lowercase letters in the same row indicate significant difference (*p* < 0.05).

**Table 3 microorganisms-10-00844-t003:** Alpha diversity of bacterial community of sainfoin silage ^1^.

Days	Treatment	Sobs	Shannon	Simpson	Ace	Chao	Good’s Coverage
Before ensiled	——	143	0.89	0.68	201.07	168.3	0.999
3	CK	852A	2.78	0.17	852A	852A	0.999
	PEG	290.33B	2.20	0.23	439.9B	433.4B	0.997
7	CK	329	2.41	0.21	456.05	470.25	0.997
	PEG	552.67	2.42	0.24	596.04	604.67	0.999
14	CK	357	2.61	0.18B	544.22	488.43	0.997
	PEG	307.67	2.20	0.26A	423.48	427.06	0.997
30	CK	410.67A	2.23	0.18B	538.13	554.1	0.999
	PEG	352.67B	2.10	0.31A	501.24	496.38	0.999
60	CK	229.33	1.88	0.31	317.79	295.08	0.999
	PEG	199.67	1.85	0.28	289.93	269.35	0.999

^1^*n* = 3. CK: control group; PEG: polyethylene glycol; Different capital letters in the same column indicate significant difference (*p* < 0.05).

**Table 4 microorganisms-10-00844-t004:** Alpha diversity of fungi community of sainfoin ensiled with or without the addition of PEG.

Days	PEG	Sobs	Shannon	Simpson	Ace	Chao	Goods Coverage
Before ensiled	N	148	2.25	0.18	192.26	197.05	0.999
3	Y	122.67B	2.97A	0.10	127.30B	125.66B	0.999
	N	169.67A	2.44B	0.13	193.21A	192.22A	0.999
7	Y	79.67B	2.84	0.10	81.87B	80.92B	0.999
	N	165.67A	2.63	0.11	175.83A	177.83A	0.999
14	Y	55.00B	2.48	0.11	58.31B	57.72B	0.999
	N	126.00A	2.57	0.18	130.69A	132.56A	0.999
30	Y	36.33B	2.69	0.09	47.63B	39.00B	0.999
	N	125.33A	2.61	0.11	129.43A	130.26A	0.999
60	Y	59.00	3.08	0.11	66.81	60.50	0.999
	N	51.00	2.95	0.08	51.98	51.61	0.999

Different capital letters in the same column indicate significant difference (*p* < 0.05). N: without addition with PEG; Y: addition with PEG.

## Data Availability

The data supporting characteristics of fermentation and wilted sainfoin can be found at Mendeley Data: https://doi.org/10.17632/rxsjb8vw6b.1 (accessed on 26 March 2022). The data supporting the bacteria and fungi community of Sequence Read Archive (SRA) can be found at National Center for Biotechnology Information (NCBI). SRA number for fungi: PRJNA811366. SRA number for bacteria: PRJNA796365 (accessed on 26 March 2020).

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
