# Peer review of "Effect of Intrinsic Tannins on the Fermentation Quality and Associated with the Bacterial and Fungal Community of Sainfoin Silage"

_microorganisms, 2022, doi:10.3390/microorganisms10050844_

Round 1

Reviewer 1 Report

The difference in dry matter suggests that one pile was treated with PEG and then pile used to fill all bags and then check pile was used to fill all bags. If so, one does not have replications but rather subsampling and not statistical analysis.  This needs to be clarified and paper not publishible if first is the case.

line 75 I presume that each day sampling was from a separate bag for a total 5*2*3 or 30 bags.  This should be specified.

Table 1 what are units of CP NDF, ADF, Neutral detergent insoluble protein, acid detergent insoluble protein and soluble protein?

Table 2 should be formated so that units are on same line as analyte.

Why is the DM of the check less than PEG treated?

Did you have one big pile that was treated with PEG and then split into three bags or did you have 3 separate piles that were treated separately and bagged randomly with check sample bagging? If former than you have subsampling, not replication.

Should be reviewed for English. Following are a few of the worst statements:

P1 L12/13  CT function includes inhibit bacterial and fungi activity during ensiling process change to CT functions include inhibition of bacterial and fungal activity during ensiling process.

P1 L28/29  The large number of agricultural scientist study on condensed tannins (CT) more than 50 years. Change to Many agricultural Scientists have studied condensed tannins (CT) over the past 50 years.

P1 L31/32  This protection can improve silage quality and increased undegraded proteins to flow into hindgut of ruminants change to This protection can improve silage quality and increase undegraded protein flow into the abomasum.

Cattle stomachs have four compartments: the rumen, the reticulum, the omasum and the abomasum (where bypass proteins are digested) and then the duodenum.

P2 L57 change clarify to clarifying

P2 L59 change fungi to fungal

P8 L205 some words seem to be missing at beginning of line.

Numerous other English issues occur during the article, please check carefully

Author Response

Response to Reviewer 1 Comments

Dear Reviewers:

Thank you for the reviewers’ comments concerning our manuscript entitled “Effect of intrinsic tannins on the fermentation quality and associated with the bacterial and fungal community of sainfoin silage”(ID:1680307). Those comments are valuable and very helpful for revising and improving our paper, as well as the important guiding significance to our researches. We have studied comment carefully and have made correction which we hope meet with approval.

Point 1: The difference in dry matter suggests that one pile was treated with PEG and then pile used to fill all bags and then check pile was used to fill all bags. If so, one does not have replications but rather subsampling and not statistical analysis.  This needs to be clarified and paper not publishible if first is the case.

Response 1: We are very sorry for our incorrect writing about ensiling methods. Actually , we made three replications for each PEG treatments. After havested of sainfoin, for PEG treatment, we random collected 1000g samples and sprayed with PEG solution for each silage bags , three replicates of silage bags for each fermentation days (3d, 7d, 14d, 30d, 60d), which total for fifteen bags of PEG treatment. The same methods for control group expect sprayed distilled water.

Point 2: Line 75 I presume that each day sampling was from a separate bag for a total 5*2*3 or 30 bags.  This should be specified.

Response 2: Thanks for you advice , you presume is right. We specified this , you can see in the article , followed as “silage samples from a separate bag for a total 3(five fermentation days) *2(two treatment) *3(replicates) of each silage were collected”.

Point 3: Table 1 what are units of CP NDF, ADF, Neutral detergent insoluble protein, acid detergent insoluble protein and soluble protein?

Response 3: We use unified units of g/ kg DM for CP NDF, ADF, Neutral detergent insoluble protein, acid detergent insoluble protein and soluble protein. We added units in Table 1.

Point 4:Table 2 should be formated so that units are on same line as analyte.

Response 4: We formated Table 2 and mede the units on same line as analyte.

Point 5: Why is the DM of the check less than PEG treated?

Response 5: For PEG treated , the PEG stabled during ensiling. We addition 52.08 mL PEG solution( 640g/L) for each treated silage, every PEG treated silage bags contain 33.33g PEG. However, we uesd distilled water for control group, which is disappear when consider of dry matter. Therefor DM of the check less than PEG treated during ensiling.

Point 6: Did you have one big pile that was treated with PEG and then split into three bags or did you have 3 separate piles that were treated separately and bagged randomly with check sample bagging? If former than you have subsampling, not replication.

Response 6: We are very sorry about our incorrect writing for ensiling methods. Actually , we prepare 3 separate piles and treated separately, bagged randomly with check sample. We added this statements in our article.

Point 7: Should be reviewed for English. Following are a few of the worst statements:

Respone 7: We are very sorry again about our incorrect writing. Thankes for you corrected for some worst statements. You corrected for the worst statements were very useful and professional. All the worst satements that you mentioned were corrected in our article.

Point 8: Numerous other English issues occur during the article, please check carefully

Respone 8: We carefully check English in our article. Formerly, we use LetPub English extension service. We added the certificate in line 566 of our article.

Special thank to you for your good and professional comments.

Reviewer 2 Report

The article entitled “Effect of intrinsic tannins on the fermentation quality and associated with the bacterial and fungi community of sainfoin silage” is well organized; however, some inadequacy of the information in the different sections of the article rendered the possibility to accept its present form.

The following points need to be corrected and improved:

  1. In the “Introduction” :

Line 31-32: Please include a reference for this statement.

Line 57-61: Please rephrase the paragraph with a clear statement of the hypothesis and the objectives of the study. The significance of the study needs to be explained clearly in this section.

  1. In the “Materials & Method”:

Line 65 – 75: Please state clearly about the silo bags preparation. How many bags were prepared totally and how was it divided for two treatments within 5-time points?

Line 114: Please mention the author’s name before reference [21] such as …et.al. [21]

Line 117-122: Please explain the statistical analysis with more information. It looks like microbial data concentrations presented as log-transformed. Please mention that it has been done before statistical analysis. Is this experiment a Complete Randomized Design?

  1. In the “Results”:

Please provide the “n=?” in all Tables.

Please explain or elaborate more on the different capital letters and different lowercase letters in Table. Why the two types of superscripts were used?

  1. In the “Discussion”:

As the addition of the PEG treated group showed the highest DM loss, the author should explain the reason for that. Was it for the leaching or the volatilization? It would be great if the author could present the volatile fatty acids (VFAs) in a separate table.

  1. In the “Conclusion”:

The author should mention a clear conclusion on the possibility of the use of PEG in the silage.

Author Response

Response to Reviewer 2 Comments

Dear Reviewers:

Thank you for the reviewers’ comments concerning our manuscript entitled “Effect of intrinsic tannins on the fermentation quality and associated with the bacterial and fungal community of sainfoin silage”(ID:1680307). Those comments are valuable and very helpful for revising and improving our paper, as well as the important guiding significance to our researches. We have studied comment carefully and have made correction which we hope meet with approval.

Point 1: In the “Introduction” :

Line 31-32: Please include a reference for this statement.

Line 57-61: Please rephrase the paragraph with a clear statement of the hypothesis and the objectives of the study. The significance of the study needs to be explained clearly in this section.

Response 1: Actually , from line 28-33 we use the one reference which is “Zeller, W.E. Activity, Purification, and Analysis of Condensed Tannins: Current State of Affairs and Future Endeavors. Crop Science 2019, 59, 886–904, doi:10.2135/cropsci2018.05.0323.”

We rephrase the line 57-56.

Point 2: In the “Materials & Method”:

Line 65 – 75: Please state clearly about the silo bags preparation. How many bags were prepared totally and how was it divided for two treatments within 5-time points?

Line 114: Please mention the author’s name before reference [21] such as …et.al. [21]

Line 117-122: Please explain the statistical analysis with more information. It looks like microbial data concentrations presented as log-transformed. Please mention that it has been done before statistical analysis. Is this experiment a Complete Randomized Design?

Response 2: Thanks for you advice. We specified silo bags preparation, you can see in the article , followed as Line 70-73“For control and PEG treated groups, prepare 15 separate piles and treated separately, bagged randomly for control and PEG treated groups, for a total 5(five fermentation days) *2(two treatment) *3(replicates) of silage for thirty silage bags.”.

Line 114: We mention the author’s name before reference.

Line 117-122: We specified the statistical analysis, and added one reference to discribe the specific methods of bacterial and fungal community analysis at Majorbio Cloud Plantfrom (line 130 in our article). We mention that microbial data concentrations presented as log-transformed in line 108 of the article. This experiment is a Complete Randomized Design.

Point 3: In the “Results”:

Please provide the “n=?” in all Tables.

Please explain or elaborate more on the different capital letters and different lowercase letters in Table. Why the two types of superscripts were used?

Response 3: We added “n=3” in all Tables. Two types of superscripts were used cause there are two factors for our trial. We use capital letters to show treatment difference and lowercase to show fermentation days difference.

Point 4: In the “Discussion”:

As the addition of the PEG treated group showed the highest DM loss, the author should explain the reason for that. Was it for the leaching or the volatilization? It would be great if the author could present the volatile fatty acids (VFAs) in a separate table.

Response 4: We formated Table 2 and mede the units on same line as analyte. We explain the reason for PEG treated group showed the highest DM loss , in line 303-305. We present the volatile fatty acids in Table 3.

Point 5: In the “Conclusion”:

The author should mention a clear conclusion on the possibility of the use of PEG in the silage.

Response 5: We mention a clear conclusion on the possibility of the use of PEG in the silage at line 429.

Special thank to you for your good and professional comments.
